# Molecular Diagnosis in Hymenoptera Allergy: Comparison of Euroline DPA-Dx and ImmunoCAP

**DOI:** 10.3390/toxins17060310

**Published:** 2025-06-19

**Authors:** Lluís Marquès, Arantza Vega, Federico de la Roca, Carmen Domínguez, Víctor Soriano-Gomis, Teresa Alfaya, Laia Ferré-Ybarz, José-María Vega, Mario Tubella, Berta Ruiz-León

**Affiliations:** 1Allergy Department, Hospital Universitari Arnau de Vilanova, Institut de Recerca Biomèdica de Lleida Fundació Dr. Pifarré, IRBLlleida, 25198 Lleida, Spain; 2Allergy Department, Hospital Universitario de Guadalajara, ARADyAL Spanish Thematic Network and Co-operative Research Centre, 19002 Guadalajara, Spain; 3Allergy Department, Clínica Creu Blanca, ClinicAL, 08034 Barcelona, Spain; 4Allergy Department, Hospital Virgen del Puerto, 10600 Plasencia, Spain; 5Allergy Department, Hospital General Universitario de Alicante, 03010 Alicante, Spain; 6Allergy Department, Hospital Universitario Fundación Alcorcón, 28922 Madrid, Spain; teresa.alfaya@salud.madrid.org; 7Allergy Department, Althaia Xarxa Assistencial Universitària de Manresa, 08243 Manresa, Spain; 8Institut de Recerca i Innovació en Ciències de la Vida i de la Salut a la Catalunya Central (IRIS-CC), 08500 Vic, Spain; 9Allergy Department, Hospital Universitario Rio Hortega, 47012 Valladolid, Spain; 10Allergy Department, Hospital HM Nou Delfos, 08023 Barcelona, Spain; 11Immunology and Allergy Department, Hospital Universitario Reina Sofía, Instituto Maimónides de Investigación Biomédica de Córdoba (IMIBIC), ARADyAL Spanish Thematic Network and Co-operative Research Centre, Instituto de Salud Carlos III, 14004 Córdoba, Spain

**Keywords:** Hymenoptera venom allergy, in vitro sIgE testing, ImmunoCAP^®^, Euroline^®^, comparison

## Abstract

The efficacy of Hymenoptera venom immunotherapy is contingent upon the accurate identification of the insect responsible for the allergic reaction. The techniques used to detect specific IgE suffer from difficulties due to the cross-reactivity between Hymenoptera venoms (false positives), diagnostic ability, and the limited availability of allergenic components (false negatives). In this study, we analyzed the discrepancies in the results obtained with Euroline^®^ DPA-Dx and ImmunoCAP^®^ in the diagnosis of allergic reactions due to Hymenoptera stings in 151 patients. The results (positive/negative) of ImmunoCAP^®^ and Euroline^®^ agreed in 77/151 (50.99%) cases; with 15/151 (9.93%) cases positive for the same insect, and 61/151 (40.4%) cases positive for multiple insects. When the results were used to decide which venom to use for immunotherapy, there was a statistically significant discrepancy for *Polistes dominula* (21.8% of cases with ImmunoCAP^®^ compared to only 8.4% with Euroline^®^). The presence of *Polistes* venom phospholipase (Pol d 1) in Euroline^®^ did not increase its ability to differentiate double sensitization to wasps. ImmunoCAP^®^ and Euroline^®^ exhibited comparable diagnostic performance in bee venom allergy. For vespid venom allergy—particularly involving *Polistes* species—ImmunoCAP^®^ appeared to show a slight diagnostic advantage, although this finding should be interpreted with caution.

## 1. Introduction

The treatment of allergic reactions due to Hymenoptera stings with specific immunotherapy offers a protective effect in more than 95% of cases [1]. To obtain such results, it is essential to make the correct immunologic diagnosis which allows the primary sensitizing venom causing the reaction, which will be used in the immunotherapy, to be identified. The appearance of multiple positive results when using the usual diagnostic techniques (intradermal tests and the determination of specific IgE (sIgE) in serum with whole venom extract) is due to the high level of immunologic cross-reactivity between the different venoms. Molecular diagnosis sometimes allows for identifying the responsible insect (primary sensitizer) and personalizing both the diagnosis and treatment [2]. Different in vitro diagnostic techniques are available which allow sIgE against allergenic compounds to be determined. ImmunoCAP^®^ (Thermo Fisher, Uppsala, Sweden), a fluorescence enzyme immunoassay, is one of the most widely used and is often employed as a reference [3]. New techniques such as Euroline^®^ DPA-Dx (Euroimmune AG, Luebeck, Germany), offer a different methodology [4]. This test, which incorporates a new allergenic component (phospholipase from Polistes venom), is a semi-quantitative multiparametric method, based on the Immunoblot technique, which requires a small quantity of serum and with which it is possible to determine sIgE against *Apis mellifera* (Am) and its allergenic components rApi m1, rApi m 2, and rApi m 10; against *Polistes dominula* (Pd), rPol d 1, and rPol d 5; against *Vespula* spp. (Vs), rVes v 1, and rVes v 5; as well as cross-reactive carbohydrate determinants (CCDs). With the quantitative single-component system of ImmunoCAP^®^, the same allergens can be determined except for rPol d 1.

Some studies have compared these two diagnostic techniques, although only one in the context of Hymenoptera allergy. This study found a good correlation with skin tests, although some differences in diagnostic ability were observed [3,4,5,6].

The objective of this multicenter prospective observational study was to analyze the discrepancies in the results of two different in vitro diagnostic techniques with the aim of identifying the insect to which patients reacted.

## 2. Results

One hundred and fifty-one patients (69.5% male) with a median age of 45 years were included in the study. A total of 11.3% were beekeepers and 89.4% had experienced systemic reactions. The clinical suspicion of the insect responsible (identification by the patient and circumstances of the sting) was Am in 39 (25.8%) and wasp in 95 (62.9%) of the cases. In the latter group, suspicions were Pd, Vs, and *Vespa* in 21 (51.2%), 15 (36.6%), and 5 (12.2%) cases, respectively. Table 1 shows the frequencies of suspected insects and the results of intradermal skin tests (IDST).

### 2.1. Discrepancies Between Techniques

The comparison of positivity (>0.35 ku/L) between ImmunoCAP^®^ and Euroline^®^ is shown in Table 2 and in Appendix A.

ImmunoCAP^®^ was negative in 4 cases versus Euroline^®^ in 2. The individual components were negative in 9 cases for ImmunoCAP^®^ versus 5 for Euroline^®^ (Figure 1).

The results (positive/negative) of ImmunoCAP^®^ and Euroline^®^ coincide in 77/151 (50.99%) cases; with 1/151 (0.66%) cases negative for both, 15/151 (9.93%) cases positive for the same insect, and 61/151 (40.4%) cases positive for multiple insects. This coincidence increases if we compare the maximum value of specific IgE (sIgE) obtained with the two tests: up to 64.24% of coincidence and 62.91% in the detection of the same insect.

Comparing the identification of vespids with the two techniques and using the maximum value of sIgE obtained, there are significant discrepancies. In the case of the identification of Vs, there are 28 matches but 43 discrepancies, with 39 cases identified only by Euroline^®^ and 4 only by ImmunoCAP^®^ (*p* < 0.0001). For Pd, there is a coincidence in 32 cases but 45 discrepancies, with 39 detected only by ImmunoCAP^®^ while for Euroline^®^ there are only 6 (*p* < 0.0001).

### 2.2. Other Comparisons

With the insect suspected by the patient. ImmunoCAP^®^ coincides with the suspicion in 20/75 (26.67%) cases: 10 Am, 4 Vs, and 6 Pd. This test allows for discriminating 9 of 54 cases (16.67%) with suspected undifferentiated vespid stings. In the case of Euroline^®^, the figures are 14/75 (18.67%) (6 Am, 5 Vs and 3 Pd) and 11 discriminations (20.37%).

With skin tests, in 8 cases, IDSTs were negative and in 14 no results were available. ImmunoCAP^®^ matched the skin test in 66/137 (48.18%) cases (3 negative, 9 Am, 7 Vs, and 8 Pd; and 39 with multiple positives). In the case of Euroline^®^, the coincidence existed in 47/137 (34.31%) (1 negative, 5 Am, 7 vs. 4 Pd; and 30 with multiple positives).

When using a venom extract for immunotherapy (Figure 2), with ImmunoCAP^®^, Am and Pd are indicated in 26 cases (21.8%) each, and Vs in 17 cases (14.3%). With Euroline^®^, the figures are 22 cases (18.5%) for Am, 10 cases (8.4%) for Pd, and 17 cases (14.3%) for Vs. The indication of a double venom immunotherapy (VIT) is performed in 32 cases (26.9%) with ImmunoCAP ^®^ versus 49 cases (41.2%) with Euroline^®^. In 18 and 21 cases, respectively, a decision cannot be made. The results obtained with ImmunoCAP^®^ and Euroline^®^ differ especially in the groups of patients to be treated with Pd extract. The statistical analysis (McNemar test) is significant for the discrepancy (*p* < 0.001).

### 2.3. Other Results

In the 104 cases presenting sensitization to Pd as established by Euroline^®^, 54 (51.9%) presented sIgE against whole Pd extract, 64 (61.6%) against Pol d 1, and 79 (75.9%) against Pol d 5. Only 9 patients were monosensitive to Pd according to this technique, with sIgE against Pol d 1 being detected in 43% of cases and against Pol d 5 in 86% of cases. Of the 15 Pol d 5-negative cases, 13 (87%) were Pol d 1-positive. This determination was only useful to distinguish between Pd and Vs in one doubly sensitized patient. It is worth highlighting that in 40 cases of 107 in which ImmunoCAP^®^ was positive for Pd, no sIgE against Pol d 5 was detected with this technique.

The detection of sIgE against Api m 5 with ImmunoCAP^®^ in patients allergic to wasps did not provide any additional information given that all were sIgE positive for vespids.

## 3. Discussion

The circumstances of the sting, the identification of the insect, the result of the IDST, and the determination of specific IgE (sIgE) against whole venom extract are all essential pieces of information to define the Hymenoptera responsible for the allergic reaction. The determination of sIgE against allergenic components is a complementary tool that helps to make further distinctions in cases where polysensitisation is present. However, in some cases, a double allergy may coexist, especially in patients with more than one reaction after different stings. The presence of sIgE to species-specific recombinant allergens (Api m 1, Api m 3, Api m 4, Api m 10, Ves v 1, Ves v 5, Pol d 1, Pol d 5) helps to differentiate the primary sensitizer, bearing in mind that in the case of vespids there is a high cross-reactivity between group 1 and group 5 [8,9]. Other tests that are used to differentiate sensitization from allergy to insect venom are CAP inhibition and the basophil activation test. The former is considered the gold standard for patients with dual venom sensitization, and the latter has been used extensively to detect the basophil’s response to hymenopteran venoms and their components [10].

Determining the diagnostic performance of a new technique such as Euroline^®^ in comparison with more commonly used techniques (ImmunoCAP^®^) helps to assess their usefulness in routine clinical practice. Such comparisons must be made in the context of clinical practice and skin tests, bearing in mind that the sIgE values obtained are not comparable [11,12,13]. Sensitivity and specificity depend on the methodology used (e.g., solid vs. liquid phase) but also on the quality of the recombinant allergens used and their properties, such as, for example, appropriate protein folding [14,15].

Euroline^®^ is a semiquantitative technique whereas ImmunoCAP^®^ is quantitative and more suitable for monitoring sIgE levels [16,17]. It allows for the use of ratios between homologous allergens and cross-reactivity [8,16], a calculation that cannot be made with a semiquantitative technique [11], although it has been used in some studies [18]. ImmunoCAP^®^ measures sIgE levels down to 0.1 kU/L: this cut-off level is useful when total IgE levels are below 30 kU/L [19]. In contrast, Euroline^®^ allows (with 400 µL of serum) for a specific multiplex test for allergy to Hymenoptera and a complete study of each patient’s allergic profile, including phospholipase of *Polistes dominula* (Pol d 1), which is not available in ImmunoCAP^®^, and includes CCD. The cost of a multiplex technique is relatively higher but the price per allergen is lower than in a technique with a single component technique. A common request in in vitro studies of allergy to Hymenoptera venom allergy in the Mediterranean region includes the determination of sIgE against three whole venom extracts (Am, Vs, Pd) and 5 allergenic components (Api m 1, Ves v 1, Ves v 5, Pol d 1, Pol d 5) and CCD. Further research is required to determine if it is cheaper to perform such tests using a multiplex technique as compared to one with a single component.

The analysis of positivity frequencies, correlation coefficients, and agreement indices between the two techniques suggests similar results for whole venom from Am, as well as for Api m 1, Api m 10, and Pol d 5. ImmunoCAP^®^ appears to perform better for Ves v 1, whereas Euroline^®^ shows better performance for Api m 2 and Ves v 5. Comparison with whole Vs or Pd venoms is hampered by the fact that these are enriched with Ag 5 in ImmunoCAP^®^ [7]. These results suggest that the ability to detect sIgE to hymenoptera venoms and their components is comparable between both techniques. However, when these findings are applied to a global assessment of results—rather than on an allergen-by-allergen basis—and to the clinical decision of selecting venom immunotherapy composition, discrepancies emerge, suggesting that ImmunoCAP^®^ has a better performance when identifying the vespid responsible for the allergic reaction.

The discrepancies between the 2 techniques focus on the detection of sIgE to wasps, and mainly to Pd venom. Although Euroline^®^ contains Pol d 1, it appears that its capability of diagnosing Pd allergy is lower than that of ImmunoCAP^®^. This lower capability may explain a greater number of double vespid sensitizations with Euroline^®^.

The presence of phospholipase in Pd (Pol d 1) in Euroline^®^ provided little extra value in the determination of double sensitizations to wasps (Vs and Pd) in our population, probably because of the high cross-reactivity between vespid phospholipases. In our population, the presence of sIgE against Pol d 1 was clearly lower than in an Italian study [18], where 84% of polysensitized and 60% of monosensitized patients exhibited this feature. This is in line with the greater presence of sIgE against Pol d 5 in our patients as compared to those in the Italian study (76% vs. 50%). Furthermore, in that study, 95% of Pol d 5-negative patients had sIgE against Pol d 1 (as compared to 87% in our study). In another study, conducted in Spain, which used a different technique and purified natural allergens, the frequency of sIgE against Pol d 1 was similar to that found in our study [8]. We are unable to quantify the influence of geographic origin on these differences. In any case, it should be highlighted that 87% of patients identified as Pol d 5-negative with Euroline^®^ were Pol d1-positive. This allergen contributes to the diagnosis, although it has not been found to be very useful for distinguishing double sensitizations to vespids in our population, as we noted above.

As was the case with Di Fraia [4], we have observed a lower ability of Euroline^®^ to detect sIgE against whole extracts as compared to individual allergens, but in our case only for vespid venom. This could be due to the greater sensitivity of ImmunoCAP^®^ in detecting low-affinity IgE antibodies and to the enrichment of whole vespid venom with antigen 5 in this technique [7].

Limitations of this study include the low number of positive results with some of the allergens (below 50%), and that in some of the determinations with ImmunoCAP^®^, we have not obtained the desired number of cases to have a statistical power >95%. Nevertheless, the results are still useful to know better the performance of Euroline^®^.

## 4. Conclusions

In their ability to identify the insect responsible for the allergic reaction, ImmunoCAP^®^ and Euroline^®^ demonstrate comparable diagnostic performance in bee venom allergy. For *Vespula* and *Polistes dominula*, however, ImmunoCAP^®^ may offer improved diagnostic sensitivity, although this observation warrants further investigation.

## 5. Materials and Methods

Adult patients with locally extensive or systemic reactions to Hymenoptera stings but who were not undergoing VIT were recruited from 10 hospitals on the Mediterranean coast and central area of Spain. Clinical variables along with the suspected culprit insect were recorded. IDSTs were performed and Am, Pd, and Vs venom, and serum samples were taken.

Specific IgE (sIgE) levels against whole Am, Pd, and Vs extracts and their molecular allergens were measured using Euroline^®^ DPA-Dx (Euroimmune AG, Luebeck, Germany) (rApi m1, rApi m 2, rApi m 10, rPol d 1, rPol d 5, rVes v 1, rVes v 5, and CCD) and ImmunoCAP^®^ (Thermo Fisher, Uppsala, Sweden)(rApi m1, rApi m 2, rApi m 10, rPol d 5, rVes v 1, rVes v 5, and CCD).

The comparison between both diagnostic techniques was conducted within the context of their routine clinical application and aimed to reflect real-life discrepancies encountered when selecting the appropriate composition for venom immunotherapy. 

The results are described using the positive or negative values of sIgE and using the maximum value of sIgE to total venom and of all the components of the same insect in each system.

In cases with positive Am and Vs, if the result of sIgE CCD is positive (>=0.35) and the results of the sIgE to the components are available, the maximum value of the components is taken into account, and the result of the total venom is excluded.

In 119 cases (those with all parameters available), a decision about which extract for VIT would be indicated was made using the data for each technique (ImmunoCAP^®^ vs. Euroline^®^ DPA-Dx), considering the following variables:The insect identified by the patient, wherever possible, especially if the patient was a beekeeper or described close paper nests.The result of skin tests.In ImmunoCAP^®^, as it is a quantitative technique, we took into consideration values of sIgE [20,21], considering that a value of >50% for one allergen as regards another with which cross-reactivity exists to be indicative as a positive result for the former and not for the latter [19].In cases of suspected allergy to Vs or Pd, the presence of sIgE against Am with no other component except Api m 5 was not considered indicative of sensitization to bees but detection of sensitization to vespid dipeptidyl peptidases (Ves v 3 and Pol d 3).The cases in which CCD was positive and sIgE was positive for Am and Vs, but with a suspicion of allergy to only one of the insects, were assessed as false positives due to cross-reactivity, and the remaining allergenic components were used to confirm the diagnosis.The cases of suspected allergy to Vs or Pd with positive sIgE for hyaluronidase (Api m 2), but with no other positive result for Am, were assessed as false positives to bees due to cross-reactivity.The Vs and Pd venoms in ImmunoCAP^®^ are enriched with antigen 5 [7].

Quantitative variables are described using quartiles (median [25th percentile, 75th percentile]) and qualitative variables using absolute and relative frequencies of all their values. The number of valid data is also included. The McNemar test was used to detect discrepancies between tests.

It was calculated that a sample size of 150 individuals would be necessary to attain a statistical power of 95%.

## Figures and Tables

**Figure 1 toxins-17-00310-f001:**
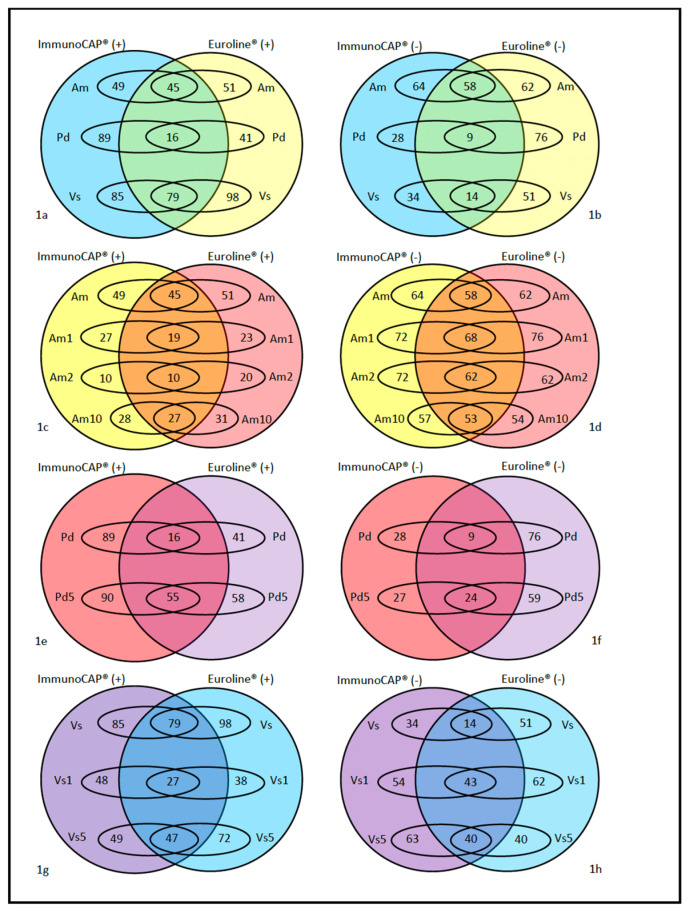
Comparison between positive (>0.35 kU/L) and negative (<0.35 kU/L) results according to ImmunoCAP^®^ and Euroline^®^ for the different venoms: (**a**,**b**) whole venom extract; (**c**,**d**): Apis mellifera and its components; (**e**,**f**): Polistes dominula and its components; and (**g**,**h**): Vespula spp. and its components. Ap: Apis mellifera. Am1: Api m 1. Am: Api m 2. Am10: Api m 10. Pd: Polistes dominula. Pd1: Pol d 1. Vs: Vespula spp. Vs1: Ves v 1. Vs5: Ves v 5.

**Figure 2 toxins-17-00310-f002:**
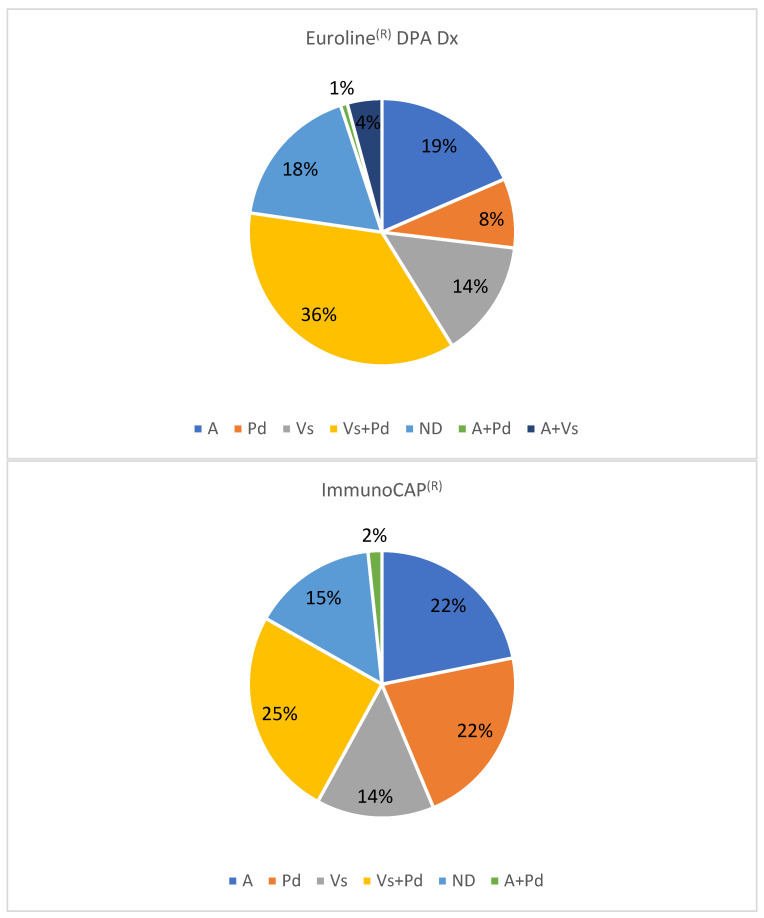
Decision of a venom extract for immunotherapy with ImmunoCAP^®^ or Euroline^®^. A: Apis mellifera. Pd: Polistes dominula. Vs: Vespula spp. ND: not defined.

**Table 1 toxins-17-00310-t001:** Clinical Suspicion and IDST Results by Insect Type. Am: *Apis mellifera*, Vs: *Vespula* spp., Pd: *Polistes dominula*. In 14 cases, IDST was not performed. NA: not applicable.

Insect	Frequency of Positive Result *n* (%)	
	Clinical Suspicion	IDST
Am	39 (25.8)	45 (34.6) (*n* = 130)
Vespid	95 (62.9)	NA
Vs	15 (9.9)	72 (55) (*n* = 131)
Pd	21 (13.9)	85 (64.9) (*n* = 131)
Vespa	5 (3.3)	NA
Vs or Pd	54 (35.8)	NA
Non identified	17 (11.2)	NA
Negative	NA	8 (5.8) (*n* = 137)

**Table 2 toxins-17-00310-t002:** Comparison of positivity between ImmunoCAP^®^ and Euroline^®^, with the observed agreement, the Kappa index of concordance with its 95% confidence interval, and the discrepancies between the methods (only >0.35 ku/L for one of the two methods, with the *p*-value if there are significant differences). N: number of patients with data from both methods. The Vs and Pd venoms in ImmunoCAP^®^ are enriched with antigen 5 [7]. Am: *Apis mellifera*, Vs: *Vespula* spp., Pd: *Polistes dominula*, CCD: cross-reactive carbohydrate determinants.

Allergen	N	Obs. Agreement	Cohen’s Kappa Index	Discrepancies (Only Euroline^®^+ vs. Only ImmunoCAP^®^+, McNemar Test *p*-Value)
**Am**	141	0.908	0.815 [0.719, 0.911]	8 vs. 5, *p* = 0.579
**Api m 1**	120	0.867	0.637 [0.477, 0.798]	5 vs. 11, *p* = 0.211
**Api m 2**	89	0.854	0.528 [0.315, 0.742]	12 vs. 1, *p* = 0.006
**Api m 10**	91	0.934	0.856 [0.745, 0.967]	5 vs. 1, *p* = 0.221
**Vs**	150	0.747	0.34 [0.177, 0.504]	27 vs. 11, *p* = 0.015
**Ves v 1**	124	0.637	0.278 [0.119, 0.437]	12 vs. 33, *p* = 0.003
**Ves v 5**	140	0.814	0.638 [0.521, 0.755]	26 vs. 0, *p* < 0.001
**Pd**	148	0.547	0.223 [0.127, 0.319]	2 vs. 65, *p* < 0.001
**Pol d 5**	139	0.799	0.598 [0.466, 0.731]	18 vs. 10, *p* = 0.186
**CCD**	56	0.714	0.269 [0.017, 0.521]	13 vs. 3, *p* = 0.024

## Data Availability

The original contributions presented in this study are included in this article and Appendix A. Further inquiries can be directed to the corresponding author.

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
