# Peer review of "Molecular Diagnosis in Hymenoptera Allergy: Comparison of Euroline DPA-Dx and ImmunoCAP"

_toxins, 2025, doi:10.3390/toxins17060310_

Round 1

Reviewer 1 Report (New Reviewer)

Comments and Suggestions for Authors

In their manuscript, the authors presented a comparison of two different platforms for detecting sIgE for Hymenoptera venoms and their respective allergens. The manuscript is relevant, as the determination of sIgE is important for diagnosis and immunotherapy choices.  However, some points should be better explained or corrected, as described below 

Major revisions

1- In table 2, present the p-values obtained for each comparison
2- Present in table 2 or as supplementary material the Immunocap data (average with S.D. of KU/L) and the class (score) obtained by the intensity of the bands after reading in the scan for each venom extract or allergen. When possible, use this data to compare the two tests using Spearman's correlation and present the p-values and the correlation coefficient. 
3- Give a possible explanation for the fact that there was a discrepancy in the analysis using the Pd extract but not in the Pol d 5 analysis when comparing the two techniques

Minor revisions 
Line 104 - change bite to sting
Table 2 - column 1 change sIgE to Allergen

Author Response

Reviewer 2 Report (New Reviewer)

Comments and Suggestions for Authors

This manuscript evaluates two in vitro diagnostic techniques, Euroline® DPA-Dx and ImmunoCAP®, for identifying the Hymenoptera venom responsible for allergies in patients. The success of Hymenoptera venom immunotherapy depends on accurately pinpointing the insect that triggers the allergic response. The study involved 151 patients and revealed considerable diagnostic inconsistencies attributed to cross-reactivity and the limited detection of allergens. The tests agreed in 50.99% of cases; however, they identified the same insect in only 9.93%. ImmunoCAP® outperformed Euroline® in identifying vespid allergies, particularly for Polistes dominula (21.8% compared to 8.4% for Euroline®). The inclusion of Pol d 1 in Euroline® did not enhance differentiation in cases of double sensitization. Both diagnostic methods yielded similar results for diagnosing bee allergies. However, certain aspects need clarification, and improvements could bring clarity and scientific rigor. After considering the following points, the manuscript can be considered for publication.

  1. Line 6: “depends on the correct identification…” could be corrected as is contingent upon the accurate identification…”
  2. Line 15: “with ImmunoCAP® and 8.4% with Euroline®” can be corrected as “21.8% with ImmunoCAP® compared to only 8.4% with Euroline®.,” which improves clarity.
  3. Line 19: “specially Polistes.” can be corrected as “especially for Polistes species.
  4. Bullet points (Lines 71–89): For better readability, consider reformatting as a clearer numbered list or paragraph.
  5. Table 1 title: Add a clear descriptor like Clinical Suspicion and IDST Results by Insect Type”
  6. Ensure consistent formatting for registered trademarks: Euroline®, ImmunoCAP®
  7. Replace all instances of “specific IgE (sIgE)” with “specific IgE (sIgE)” at first mention in each major section.
  8. Correct spacing around percent signs (e.g., “21.8 %” → “21.8%”).
  9. Consistently use “vs” or “versus” (e.g., “Euroline® vs ImmunoCAP®”).
  10. Ensure all references follow a consistent citation format (some DOIs and journal names are inconsistently styled) and confirm all citations match the in-text references.

Author Response

Reviewer 3 Report (New Reviewer)

Comments and Suggestions for Authors

This manuscript suffers from several significant shortcomings:

  • it requires extensive language editing to address grammatical and structural problems that currently impede comprehension. Significant revision is needed to clarify the authors' key points to ensure the proper scientific communication.
  • The current introduction fails to adequately situate the work within the existing research  landscape (e.g., ref 6). 
  • Experinental designs require more rigorous justification.
  • Results interpretation would benefit from more conservative claims.
Comments on the Quality of English Language

The manuscript requires extensive language editing 

Round 2

Reviewer 1 Report (New Reviewer)

Comments and Suggestions for Authors

The authors answered all the questions and provided the data as recommended. Just one point to improve the discussion based on the data presented.

Please discuss the differences when comparing the frequency of positive results and the average class score presented in table S1, especially in cases where there was a strong correlation between both tests.

Minor review

lines 200 – 201: change  ku/l to kU/L

In table S1, switch to dots instead of commas in the Spearman's correlation coefficient

Author Response

Reviewer 3 Report (New Reviewer)

Comments and Suggestions for Authors

I cannot see a great improvement of the quality of language. About the discussion, I prefer a more conservative claim, which would better serve the manuscript's stated purpose while maintaining scientific rigor. As the authors mentioned, their objective is simply comparing the two exist tests.  The absence of a comprehensive discussion making the work appears neither a comprehensive mechanistic study nor a decisive methodological advancement.

Round 3

Reviewer 3 Report (New Reviewer)

Comments and Suggestions for Authors

Looks much better now.

This manuscript is a resubmission of an earlier submission. The following is a list of the peer review reports and author responses from that submission.

Round 1

Reviewer 1 Report

Comments and Suggestions for Authors

Minor concerns

1-       In INTRODUCTION

ImmunoCAP®, a fluorescence/enzyme-im- 35 munoassay, is one of the most widely used and is often employed as a gold standard.

Comment: please put reference

2-    In ABSTRACT....In this study we compared the diagnostic efficacy of Euroline® DPA-Dx with ImmunoCAP® in the diagnosis of allergic reactions due to hymenoptera stings in 152 patients.

In RESULTS...One hundred and fifty-one patients (72% male) with a median age of 45 years were included in the study

3.2. Comparison of the attribution of the insect responsible for the reaction

Of the 152 serum samples collected.....

Comment....Please explain or correct

 3-       According to Instructions for Authors (Toxins)

References should be described with Abbreviated Journal Name.

Comment: In the REFERENCE item, please chec references number 2, 4,5,6, 7, 11 and 18

In references 16 and 17, please put et al.

Author Response

  1. In INTRODUCTION

ImmunoCAP®, a fluorescence/enzyme-im- 35 munoassay, is one of the most widely used and is often employed as a gold standard.

Comment: please put reference:

A reference (3) was added: Berge, M.; Bertilsson, L.; Hultgren, O.; Hugosson, S.; Saber, A. Qualitative and Quantitative Comparison of Allergen Component-Specific to Birch and Grass Analyzed by ImmunoCAP Assay and Euroline Immunoblot Test. European Ann Allergy Clin Immunol 2022, doi:10.23822/eurannaci.1764-1489.241.

  1. In ABSTRACT....In this study we compared the diagnostic efficacy of Euroline® DPA-Dx with ImmunoCAP® in the diagnosis of allergic reactions due to hymenoptera stings in 152 patients.

In RESULTS...One hundred and fifty-one patients (72% male) with a median age of 45 years were included in the study

3.2. Comparison of the attribution of the insect responsible for the reaction

Of the 152 serum samples collected.....

Comment....Please explain or correct 152 patients

It was a typing error. The correct value is 152, and the correction is done.

  1. According to Instructions for Authors (Toxins)

References should be described with Abbreviated Journal Name.

Comment: In the REFERENCE item, please chec references number 2, 4,5,6, 7, 11 and 18

In references 16 and 17, please put et al.

The corrections are done. Some involuntary changes could be done to the Smart Cite program used (it modifies some cites to the wrong when they are actualized). In the number of authors, we follow the norms from Toxins: (“For documents co-authored by a large number of persons (more than 10 authors), you can either cite all authors, or cite the first ten authors, then add a semicolon and add ‘et al.’ at the end:”

Reviewer 2 Report

Comments and Suggestions for Authors

Dear authors, the work would be interesting, but the purpose of the work should be formulated differently. It is very important to try to identify the insect to which the patient reacted.

Notes

1. the gold diagnostic standard is used several times and in different places it means something else: once history, skin tests, once history, skin rests and antibodies with ImmunoCap

2. the terms sensitisation and reaction are mixed up, allergy to venom does not mean anything, that's why we don't do population studies

3unicap sometimes helps? very often identifies a suspicious insect

4. the method is described illegibly

5. there are 151 patients followed by 152 sera elsewhere?

6. double allergy does not matter, what matters is whether the patient reacts to 2 insects and this means double allergy

7. no description of recombinants of wasp venom or bees, which do not have a CCD and indicate a primary sensitization to a given venom

8. no restrictions in work and there are a lot of methods, group size and one way of calculations, after all, to what do we compare? for skin testing and interview? impossible - skin tests are not very specific, they often do not identify the insect, so it was a challenge to compare EUROIMMUN TO UNICAP100

9. conclusions are wrong

10 compliance below 50% is not acceptable! it's like roulette

11. positive sIgE result above 0.35 according to what literature??? no citation

Author Response

Dear authors, the work would be interesting, but the purpose of the work should be formulated differently. It is very important to try to identify the insect to which the patient reacted.

We have changed the introduction as to improve the explanation of the purpose of the work.

Notes

  1. the gold diagnostic standard is used several times and in different places it means something else: once history, skin tests, once history, skin rests and antibodies with ImmunoCap

The text states in different points (eg, L75, L172), that the gold standard used in this investigation as a reference for identifying a culprit insect was the clinical history plus skin tests.

  1. the terms sensitisation and reaction are mixed up, allergy to venom does not mean anything, that's why we don't do population studies

In the introduction there is a differentiation between sensitization and attribution of responsible insect. For improving this aspect some changes are done (L32 and L218). The concepts that we have used are: sensitization means detection of specific IgE to an allergen, allergy means clinical reaction with this allergen. In the end this is the reason of the work: to know the utility of Euroline in helping differentiate the two things, comparing with the techniques that are in use in everyday practice.

  1. unicap sometimes helps? very often identifies a suspicious insect

The study used ImmunoCAP as a comparator of Euroline. In the results we show the capabilty of identifying the insect in our population with this technique. It was not the aim of the study to review ImmunoCAP in hymenoptera allergy. In cites num 2,7 or 8 is possible to obtain further information.

  1. the method is described illegibly.

We have corrected the description of the methodology used, to make it more clear to the readers.

  1. there are 151 patients followed by 152 sera elsewhere?

152 patients, it was a typing error and it was corrected.

  1. double allergy does not matter, what matters is whether the patient reacts to 2 insects and this means double allergy

      We have included a new reference in the bibliography (13) that explains in an algorithm the diagnostic procedure in hymenoptera allergy, following the european guidelines of 2005 (this is the last time they were published; the cited reference explains that they are in process of actualization). In the introduction we explain the importance of the identification of the insect that caused the reaction, to which the patient is allergic and a short text on this matter is included in the discussion.

  1. no description of recombinants of wasp venom or bees, which do not have a CCD and indicate a primary sensitization to a given venom

All the recombinant allergens analyzed are those present in the two techniques and are recognized as primary sensitizers to wasps and bees. Due to the restriction in the extension of the text we had not included basic concepts in molecular allergology of venoms, but references are given (2 and 9)

  1. no restrictions in work and there are a lot of methods, group size and one way of calculations, after all, to what do we compare? for skin testing and interview? impossible - skin tests are not very specific, they often do not identify the insect, so it was a challenge to compare EUROIMMUN TO UNICAP100

Following the current guidelines (EAACI 2005), the diagnosis of insect venom allergy is done, as in other allergic diseases, with clinical history and detection of specific IgE, with skin tests and/or in vitro tests. The ideal, but uncommon, situation is when the patient brings the insect. A recent algorithm (cite 13) states that skin tests and/or determination of serum IgE could be employed. It is not accepted to perform sting challenges in patients untreated, in a different way as challenge tests are used in diagnosing food or drug allergy for example. The study follows the ordinary clinical procedure and intends to see what happens when one or another in vitro technique is used.

  1. conclusions are wrong.

We think that the conclusions have been obtained from the results obtained, and the differences between the two tests are minor. The limitations of this study and it’s analysis do not invalidate the results, and the two techniques (been conscious that no perfect test exists) could be used equally in everyday practice.

  1. compliance below 50% is not acceptable! it's like roulette.

The values below 50% are a limitation of our study and we have included a commentary in the discussion.

  1. positive sIgE result above 0.35 according to what literature??? no citation

We have included a citation on this matter: 7. 

Reviewer 3 Report

Comments and Suggestions for Authors

1       Introduction – “offer new allergenic components and a different methodology [3]. The new test is a semi-quantitative multiparametric method, based on the Immunoblot technique, which requires a small quantity of serum and with which it is possible to determine sIgE against Apis mellifera (Am) and its allergenic components rApi m1, rApi m 2 and rApi m 10; against Polistes dominula (Pd), rPol d 1 and rPol d 5; against Vespula vulgaris (Vv), rVes v 1 and rVes v 5; as well as cross-reactive carbohydrate determinants (CCD). With the quantitative single component system of ImmunoCAP® the same allergens can be determined with the ex-44 ception of rPol d 1.” – it does not make sense – first, you claim that there  are “new allergenic components” allowed to be determined, but later you claim that only Pold1 is newly included. Please clarify.

2       L.58 – “Intradermal skin tests were performed with Am, Pd and Vv venom” – how did you ensure that there were no stings by Vespula germanica?

3       Table 1 – it appears that at least some patients were tested only with one type of the test? Specify how many patients were tested by both tests for each allergen type.

4       L.103 – “was assessed using the Kappa 103 coefficient measure (with statistical significance set at P <0.05).” – the p values were not provided in the Results.

5       L.105 – “It was calculated that a sample size of 150 individuals would be necessary to attain a statistical power of 95%.” – it is unclear why the sample sizes were mostly below 150 in the majority of the tests performed (see Table 1)

6       L.106 – “was approved by the Ethics Committee” – be more specific on which Ethics Committee approved the study protocol, when was it approved, and what was the number of the approval.

7       L.106 – “and informed consent was” – was the informed consent written?

8       Table 1 – “Frequency of positive results” – how did you define a “positive” result?

9       L.118 – “ImmunoCAP® was negative in 4 cases and Euroline® in 2.” – some Venn diagram (or other graphics) should be shown, combining the clinical suspicion and test outcomes.

10    The statistical analyses should be improved. Because multiple parameters were tracked, the results should be analyzed using some multiparametric approach.

11    Other alternatives to the two tested methods should be mentioned. These include (but are not limited to) the basophil activation test, which was repeatedly used in the detection of the response of basophils to the hymenopteran venoms and their components (check and cite, e.g., Clin Exp Allergy. 2019 Jan;49(1):54-67, Mol Immunol. 2022 Sep;149:59-65, and J Allergy Clin Immunol Pract. 2020 Jan;8(1):392-394).

Author Response

  • Introduction – “offer new allergenic components and a different methodology [3]. The new test is a semi-quantitative multiparametric method, based on the Immunoblot technique, which requires a small quantity of serum and with which it is possible to determine sIgE against Apis mellifera(Am) and its allergenic components rApi m1, rApi m 2 and rApi m 10; against Polistes dominula (Pd), rPol d 1 and rPol d 5; against Vespula vulgaris (Vv), rVes v 1 and rVes v 5; as well as cross-reactive carbohydrate determinants (CCD). With the quantitative single component system of ImmunoCAP® the same allergens can be determined with the ex-44 ception of rPol d 1.” – it does not make sense – first, you claim that there  are “new allergenic components” allowed to be determined, but later you claim that only Pold1 is newly included. Please clarify.

            We have changed the text in the introduction.

  • 58 – “Intradermal skin tests were performed with Am, Pd and Vv venom” – how did you ensure that there were no stings by Vespula germanica?

Is quite impossible for the stung patient to distinguish between the different species of vespula. The allergenic composition of the venoms from the Vespinae sub-family is very similar and a high degree of cross-reactivity exists between the different genera (Vespula, Vespa, Dolichovespula). Among the venoms from the insects of the Vespula genera the best characterized from the allergological point of view is that from Vespula vulgaris, with some characterized allergens available for molecular diagnosis. Different allergens from V. vulgaris show sequence homology with that of V. germanica.

In the situation he or she could bring the responsible insect, and could be identified by an entomologist, if it is of the genera vespula (germanica or vulgaris), there will be no changes in the extracts used for diagnosis or treatment. (cite 14). Other papers reflect the same routine (cite 16, suggested in comment num 11).

The extracts for intradermal tests used in our patients included a mix of V. germanica + V. vulgaris (Roxall) or a mix of V. germanica and american species (V alascensis, V maculifrons, V flavopilosa, V pensylvanica and V squamosa)(ALK), defined as Vespula spp.

We had wrote “Vv” with the pretension of unifying and simplifying things and abbreviatures, but we accept the correction and have made changes in the text, modifying it to Vespula species (Vs).

3       Table 1 – it appears that at least some patients were tested only with one type of the test? Specify how many patients were tested by both tests for each allergen type.

   Yes, there were some patients that were not tested to all components in ImmunoCAP. This is due that, in some hospitals, restrictions exist in the number of determinations that could be done (e.g., in cases with a clear suspicion of sting of a wasp, the determinations to bee venom were not performed).

  • 103 – “was assessed using the Kappa 103 coefficient measure (with statistical significance set at P <0.05).” – the p values were not provided in the Results.

In all cases, and with the SPSS software used, all the values of P appear as: 0.000, not including the SPSS version the 4th decimal in this output of results. We can say that all P-values are less than 0.001. This result appears in the description of the figure 1.

5       L.105 – “It was calculated that a sample size of 150 individuals would be necessary to attain a statistical power of 95%.” – it is unclear why the sample sizes were mostly below 150 in the majority of the tests performed (see Table 1).

As we have commented in point num 3, some determinations were not done. Another point to take into account is that is impossible that all 152 serums presented with IgE to all the allergens, as there exists different profiles of IgE in every patient. But this is a limitation of our study, and by this reason the statistical power will be lower than the 95% desired. The text of the discussion has included this point.

6       L.106 – “was approved by the Ethics Committee” – be more specific on which Ethics Committee approved the study protocol, when was it approved, and what was the number of the approval.

The information was removed by editor for double blind peer review. A copy of the approbation was sent to the editor.

7       L.106 – “and informed consent was” – was the informed consent written?

The information was removed by editor for double blind peer review. A copy of the informed consent form was sent to the editor.

8       Table 1 – “Frequency of positive results” – how did you define a “positive” result?

In the methods we state, “Any value over 0.35 kU/L was considered positive”. We have included this information in the table.

9       L.118 – “ImmunoCAP® was negative in 4 cases and Euroline® in 2.” – some Venn diagram (or other graphics) should be shown, combining the clinical suspicion and test outcomes.

A diagram is done to show the different results for every technique. We apologize but due to time and space constrictions it has been impossible to include more results.

10    The statistical analyses should be improved. Because multiple parameters were tracked, the results should be analyzed using some multiparametric approach.

For a kappa analysis of two diagnostic tests, it is not necessary to consider other confounding factors, it is not a multivariate analysis. The results could have been stratified by sex, age... but that was not the objective of the study and we do not believe that doing so is justified. We have followed the kind of analysis done in similar studies comparing techniques (cites 4,5 and 6).

11    Other alternatives to the two tested methods should be mentioned. These include (but are not limited to) the basophil activation test, which was repeatedly used in the detection of the response of basophils to the hymenopteran venoms and their components (check and cite, e.g., Clin Exp Allergy. 2019 Jan;49(1):54-67, Mol Immunol. 2022 Sep;149:59-65, and J Allergy Clin Immunol Pract. 2020 Jan;8(1):392-394).

A mention to the two techniques and a cite (16) are included (L194-197)

Round 2

Reviewer 2 Report

Comments and Suggestions for Authors

Dear Authors, there are still major logical errors

1 recognition of what insect triggered the reaction is based on: history, skin test and / or sIgE, [Guidelines EAACI 2017 ], which puts in vivo and in vitro tests on an equal footing, so the very assumption of the gold standard for diagnosing insect venom allergy without sIgE is today  a mistake. For skin tests, we use natural extracts containing CCD, and therefore often skin tests giving false positive results - methodological error of the whole work! Recombinant allergens without a CCD can mostly indicate which insect was the cause of the reaction.

2. the purpose of the work is to indicate what insect was the cause of the reaction: there is no answer to this question in the final conclusions, there are other conclusions, not resulting from the work

3. again, there is no explanation and no distinction between double sensitization, which has no clinical significance, and the occurrence of 2 x reactions after being stung by different insects, so there is no information on how many people reacted more than once after being stung

4. analyzed 123 cases, 60% of them respond to Uni Cap markings, and 43% to Euro DAX, this is a major difference, and what next? please explain graphically, tabularly what was done next, in 20 people 2 insects - clinically 2 x reaction after a different insect did the results of additional tests indicate ??? what about the rest people?

? why were people with a hornet reaction kicked out? we perform VIT with wasp venom if there is no vaccine with hornet venom and diagnose using skin tests with wasp venom or sIgE for hornet venom - there is very important  threat to the patient's 

5 agreement factor 0.36... is marginal in L 164 - agreed, but elsewhere for Euro  DAX and UniCAP  for Pd - allows for practical use - unacceptable?

6. Conclusions in this form are a not true. Someone who does not read the whole work, and only the final conclusion will be able to choose cheaper and less sensitive Euro tests and feel safe with a negative result, and if the patient develops anaphylaxis, the authors of this work will have the patient and the doctor who read the article on their conscience

7. apart from the lack of accuracy of the methodology, the conclusion is one Euro is suitable for diagnosing and confirming allergies to bee venom, while wasps: wasp, dolichovespula - cannot be diagnosed using the DAX test

8. are there numbers in fig 2? Whether % ; please unify Vs and not Vv

Author Response

Dear Authors, there are still major logical errors

  1. recognition of what insect triggered the reaction is based on: history, skin test and / or sIgE, [Guidelines EAACI 2017 ], which puts in vivo and in vitro tests on an equal footing, so the very assumption of the gold standard for diagnosing insect venom allergy without sIgE is today  a mistake. For skin tests, we use natural extracts containing CCD, and therefore often skin tests giving false positive results - methodological error of the whole work! Recombinant allergens without a CCD can mostly indicate which insect was the cause of the reaction.

Our aim was to compare the efficacy in the diagnosis of hymenoptera allergy with Euroimmune compared to ImmunoCAP (as the historical reference). In this case we are analyzing frequency of sensitization, not allergy. This is reflected in the comparison of the frequency of positive results and the kappa analysis.

The other way to analyze its diagnostic performance was to compare the results of the decision (with all the limits it has, but as it is used in everyday practice following international guidelines) about the culprit insect considering clinical history, skin tests and in vitro tests (with ImmunoCAP or with Euroline). We are using sIgE for the diagnostic process, comparing the two techniques. We are not saying that the gold standard in hymenoptera allergy diagnosis is clinical history and skin tests: we say that we are using it as the comparator or reference (and have used the term of gold standard in this sense). To avoid this confusion, we have eliminated the terminology gold standard and changed it for “reference”.

We have analyzed different comparative studies before deciding the methodology to use in our study and the method to define sensitization or allergy to hymenoptera venom is clinical history, intradermal tests and/or IgE determination.

Schrautzer. Sensitivity and specificity of Hymenoptera allergen components depend on the diagnostic assay employed. JACI  2016;137(5):1603–1605. First, we aimed to compare the sensitivity of rApi m 1 between the CAP and Immulite systems. We therefore included 111 bee venom–allergic patients from Graz and 55 patients from another center in Salzburg. All included patients have had experienced systemic sting reactions. Hymenoptera venom allergy was confirmed by intradermal testing, and/or IgE determination (CAP), and/or the basophil activation test. Intradermal tests were performed with 0.02 mL of 0.01, 0.1, and 1 mg/mL using purified bee and vespid venom extracts (ALK-Abell_o, Hørsholm, Denmark). The test result was positive if a wheal of 5 mm or more in diameter and a concomitant erythema occurred. IgE determination was uniformly done for all centers in Graz. Values greater than 0.35 kU/L in the CAP system and 0.34 kU/L in the Immulite system were considered positive, respectively.

  1. the purpose of the work is to indicate what insect was the cause of the reaction: there is no answer to this question in the final conclusions, there are other conclusions, not resulting from the work

We have made changes in abstract and conclusions:

In their ability to identify the insect responsible for the allergic reaction, ImmunoCAP® and Euroline® offer a similar diagnostic performance for allergy to bees. Instead, in the case of Vs and Pd, the diagnostic efficacy of ImmunoCAP® is better than Euroline®. Agreement between the two techniques is very high for Am, good to moderate for Vs, and moderate to weak for Pd.

  1. again, there is no explanation and no distinction between double sensitization, which has no clinical significance, and the occurrence of 2 x reactions after being stung by different insects, so there is no information on how many people reacted more than once after being stung

Effectively, there was no explanation differentiating double sensitization and double allergy, that could be present in patients with more than one reaction. For this reason, some changes have been made (L112, L252-256). In the comparison of two diagnostic techniques, the relevance of reacting more than once could be marginal and will affect equally the two methods.

  1. analyzed 123 cases, 60% of them respond to Uni Cap markings, and 43% to Euro DAX, this is a major difference, and what next? please explain graphically, tabularly what was done next, in 20 people 2 insects - clinically 2 x reaction after a different insect did the results of additional tests indicate ??? what about the rest people?

We don’t understand the question/comment.

? why were people with a hornet reaction kicked out? we perform VIT with wasp venom if there is no vaccine with hornet venom and diagnose using skin tests with wasp venom or sIgE for hornet venom - there is very important  threat to the patient's 

Because the number of cases was too low to draw any conclusions and because the profile of specific IgE in these patients could alter the results. We thought precautionary not to analyze these results.

  1. agreement factor 0.36... is marginal in L 164 - agreed, but elsewhere for Euro  DAX and UniCAP  for Pd - allows for practical use - unacceptable?

We have changed, as told before, the conclusions in this aspect. We do not think, with only our results, that this qualification could be used. The results could change in different populations owing, for example, to different profiles of sIgE (Pol d 1 vs Pol d 5).

  1. Conclusions in this form are a not true. Someone who does not read the whole work, and only the final conclusion will be able to choose cheaper and less sensitive Euro tests and feel safe with a negative result, and if the patient develops anaphylaxis, the authors of this work will have the patient and the doctor who read the article on their conscience

We have changed, as told before, the conclusions.

  1. apart from the lack of accuracy of the methodology, the conclusion is one Euro is suitable for diagnosing and confirming allergies to bee venom, while wasps: wasp, dolichovespula - cannot be diagnosed using the DAX test

refer to the comments 5 and 6

  1. are there numbers in fig 2? Whether % ; please unify Vs and not Vv

Vv has been changed to Vs

Reviewer 3 Report

Comments and Suggestions for Authors

1)      “spp.” in “Vespula spp.” should not be italicized.

2)      In the rebuttal letter, you claim that “restrictions exist in the number of determinations that could be done” – specify how many patients were tested with multiple tests and provide test outcomes for this group analyzed separately  - similarly as it was already done in Fig. 2.

3)      Review question #9 – “We apologize but due to time and space constrictions it has been impossible to include more results.” – there are no time and space constrictions which would prevent the reflection of this point. If more time is needed, it can be requested or the manuscript can be resubmitted when the data are ready.

4)      Review question #10 – “For a kappa analysis of two diagnostic tests, it is not necessary to consider other confounding factors, it is not a multivariate analysis. The results could have been stratified by sex, age... but that was not the objective of the study and we do not believe that doing so is justified. ” – I understand that the currently disclosed analyses were not multiparametric. That is why the multiparametric analyses were requested to be added. There are multiple potential confounding factors, you have data for some of them, therefore, a multiparametric approach would be more appropriate than the simple correlations.

5)      Review question #11 – the comment was incompletely reflected. Why should BAT be considered a “promising technique” only? It is already in routine use for one or two decades in many countries over the world. Adhere more strictly to the raised comment.

Author Response

1)      “spp.” in “Vespula spp.” should not be italicized. CORRECTED

2)      In the rebuttal letter, you claim that “restrictions exist in the number of determinations that could be done” – specify how many patients were tested with multiple tests and provide test outcomes for this group analyzed separately  - similarly as it was already done in Fig. 2.

                  The patients tested with “multiple tests” (we understand Euroline and ImmunoCAP already) are those collected in table 1. The statistical analysis comparing the sensitizations to the different allergens or the attribution of the responsible insect is done with this group.

3)      Review question #9 – “We apologize but due to time and space constrictions it has been impossible to include more results.” – there are no time and space constrictions which would prevent the reflection of this point. If more time is needed, it can be requested or the manuscript can be resubmitted when the data are ready.

                  The editor gave us a limited number of days to correct the manuscript, request already done.

4)      Review question #10 – “For a kappa analysis of two diagnostic tests, it is not necessary to consider other confounding factors, it is not a multivariate analysis. The results could have been stratified by sex, age... but that was not the objective of the study and we do not believe that doing so is justified. ” – I understand that the currently disclosed analyses were not multiparametric. That is why the multiparametric analyses were requested to be added. There are multiple potential confounding factors, you have data for some of them, therefore, a multiparametric approach would be more appropriate than the simple correlations.

Doing a multiparameter analysis is out of our purpose in this comparative study. We do not intend in it to delve into those confounding factors that we think would affect the two techniques equally. Of course, many analysis could be done with all the data we gathered, but it was not the purpose of this paper.

5)      Review question #11 – the comment was incompletely reflected. Why should BAT be considered a “promising technique” only? It is already in routine use for one or two decades in many countries over the world. Adhere more strictly to the raised comment.

We have used the appreciation done in the bibliographic cite 16 (This overview will not examine cell-based assays such as the basophil activation test (BAT) as discussed elsewhere because the BAT and associated basophil assays remain important research-based assays that are performed in limited immunology laboratories, but rarely used as routine diagnostic tests.). But we have changed the text.